# Development of New Iso-Cytoplasmic Rice-Restorer Lines and New Rice Hybrids with Superior Grain Yield and Grain Quality Characteristics by Utilizing Restorers’ Fertility Genes

**DOI:** 10.3390/genes13050808

**Published:** 2022-05-01

**Authors:** Mamdouh M. A. Awad-Allah, Kotb A. Attia, Ahmad Alsayed Omar, Eldessoky S. Dessoky, Fahad Mohammed Alzuaibr, Mohammed Ali Alshehri, Mohamed A. Abdein, Azza H. Mohamed

**Affiliations:** 1Rice Research Department, Field Crops Research Institute, Agricultural Research Center, Giza 12619, Egypt; kattia1.c@ksu.edu.sa; 2Department of Biochemistry, College of Science, King Saud University, Riyadh 11451, Saudi Arabia; 3Biochemistry Department, Faculty of Agriculture, Zagazig University, Zagazig 44519, Egypt; omar71@ufl.edu; 4Citrus Research & Education Center, Institute of Food and Agricultural Sciences (IFAS), University of Florida, 700 Experiment Station Road, Lake Alfred, FL 33850, USA; azza@ufl.edu; 5Department of Biology, College of Science, Taif University, Taif 21944, Saudi Arabia; es.dessouky@tu.edu.sa; 6Biology Department, College of Science, Tabuk University, Tabuk 71491, Saudi Arabia; falzuaiber@ut.edu.sa (F.M.A.); Ma.alshehri@ut.edu.sa (M.A.A.); 7Biology Department, Faculty of Arts and Science, Northern Border University, Rafha 91911, Saudi Arabia; 8Department of Agricultural Chemistry, College of Agriculture, Mansoura University, Mansoura 35516, Egypt

**Keywords:** newly developed restorer lines, grain yield, yield components, heterosis, combining ability, *Oryza sativa* L.

## Abstract

This research was carried out at the Experimental Farm of Sakha Agricultural Research Station, Sakha, Kafr El-Sheikh, Egypt, during the 2018–2020 rice-growing seasons to develop and evaluate four iso-cytoplasmic rice-restorer lines: NRL79, NRL80, NRL81, and NRL82, as well as Giza 178, with ten new hybrids in order to estimate genotypic coefficient, phenotypic coefficient, heritability in a broad sense, and advantage over Giza 178 as a check variety (control) of new restorer lines. This study also estimated combining ability, gene action, better-parent heterosis (BP), mid-parents heterosis (MP), and standard heterosis (SH) over Egyptian Hybrid one (IR69A × Giza 178) as a check hybrid (control) for grain yield, agronomic traits, and some grain quality characters in restorer lines and hybrids. The percentage of advantage over commercial-variety Giza 178 (check) was significant, and highly significant among the newly developed restorer fertility lines for all the studied traits. This indicates that the selection is a highly effective factor in improving these traits. New restorer fertility lines showed highly significant positive values over commercial restorer for grain yield; the values ranged from 51% for NRL80 to 100.4% for NRL82, respectively. Meanwhile, in regard to the grain shape of paddy rice, three lines of the promising lines showed highly significant negative desirable values compared with Giza 178; the values ranged from −7.7% for the NRL80 to −15.2% for NRL79, respectively. Based on the superiority of the new lines, the new lines can be used as new restorer fertility lines to breed promising new hybrids and new inbred rice lines or varieties. From the results of the testcross experiment, the four promising lines were identified as effective restorer fertility lines for two cytoplasmic male sterile (CMS) lines. Moreover, the six rice hybrids showed values for SH heterosis of grain yield/plant of more than 15% over the check hybrid variety, with high values of 1000-grain weight and desirable grain shape; these hybrids were G46A × NRL81 (125.1%), G46A × NRL80 (66.9%), IR69A × NRL79 (47.2%), G46A × NRL79 (24.6%), IR69A × NRL81 (23.4%), and IR69A × NRL82 (16.2%).

## 1. Introduction

Rice is the main food-grain crop for more than half of the world’s population. The use of heterosis in rice was taken advantage of and had an important role in increasing the rice yield after the first green revolution [1]. To meet the increase in population, we will have to produce 40% more rice by 2030 to satisfy the growing demand without adversely affecting the source base adversely. This increased demand will have to be met with less land, less water, less effort, and fewer chemical fertilizers. To produce more rice from available suitable land to meet the increased demand for rice, we need high-yielding rice varieties with greater yield stability. To increase the yield potential of rice, various strategies are being employed. These strategies include hybrid rice breeding, producing 15–20% higher grain yield than the best varieties [2], which thus will help overcome the yield gap and meet the challenge of increasing and preserving rice production from the same available natural resources [3]. Most of the hybrid rice varieties produced in many countries of the world are based on the cytoplasmic genetic male sterility (CGMS) system [4]. It is known as the three-line system because it consists of three parental lines that include hybridization between the three parental lines [5]. Hybrid rice breeding based on the CGMS system in developing new hybrids in rice is probable, only when effective restorer lines are available [6]. The low frequency of ideal maintainers and restorer lines among elite breeding lines is considered the biggest limiting factor in hybrid rice breeding. Consequently, to develop heterotic hybrids as well as to improve breeding efficiency, breeding to improve parental lines is essential in hybrid rice-breeding programs [7].

To increase the efficiency of hybrid rice-seed production, it is necessary to improve the outcrossing rate of CMS lines [8]. The percentage of seed set in a CMS line of hybrid rice-seed production depends upon the extent of outcrossing, which is a function of floral morphology and flowering behavior of CMS lines and restorer lines [9].

To make progress and to meet the expanding prospects of hybrid rice improvement, it is necessary to collect, evaluate, improve, and maintain parent lines. A selection of segregation generation from the promising rice hybrid is considered a novel approach and an effective method to develop new iso-cytoplasmic restorer lines. The newly developed iso-cytoplasmic restorer lines have the same source as the cytoplasmic male sterility, which could minimize the harmful interaction between the cytoplasmic and nuclear genes [10]. The parent-selection method for hybridization is considered one of the essential factors in order to obtain a successful breeding program [11].

The objective of this study was designated for developing new iso-cytoplasmic restorer fertility lines from commercial hybrids and promising rice hybrids grown in Egypt, as well as their evaluation to identify promising iso-cytoplasmic rice-restorer fertility lines that can further be utilized in the development of improved rice hybrids under Egyptian conditions.

## 2. Materials and Methods

### 2.1. Genetic Materials

This study was carried out at the Experimental Farm of Sakha Agricultural Research Station, Sakha, Kafr El-Sheikh, Egypt, during the 2015–2020 rice-growing seasons. The materials experimental in this study consisted of 57 iso-cytoplasmic restorer fertility lines derived from a commercial hybrid, namely Egyptian Hybrid one (IR69625A × Giza 178), and a promising rice hybrid (G46A × BG34-8). The materials were evaluated for floral traits, some grain quality traits, grain yield, and yield contributing traits. The genotype’s names and the parentage of the materials studied are presented in Table 1.

### 2.2. Field Evaluation

In the first step, F_1_ seeds were produced by hand crossing 2012; in the next year, pure seeds of two rice hybrids were grown and F_2_ seeds were harvested on a single-plant basis. These F_2_ seeds and subsequent generations were grown in the field and subjected to generation advancement coupled with selection for phenotypic performance and grain yield per plant, spikelet fertility, number of days to 50% flowering, panicles per plant, plant height, panicle excretion, anther length (AL), anther breadth (AB), and filament length (FL) (Figure 1). During rice season 2015, a set of 57 iso-cytoplasmic restorer lines were generated from the two hybrids. The crop was grown following recommended agronomic practices and plant-protection measures to ensure proper crop growth. Restorer lines of 57 iso-cytoplasmic populations along with the parental lines were evaluated for phenotypic performance and yielding ability. Out of 57 iso-cytoplasmic restorer lines, the best 23 lines, selected based on their phenotype and yield ability, were hybridized with two cytoplasmic male sterile lines. Out of 23 iso-cytoplasmic restorer fertility lines, the best 4 lines based on their phenotype, grain yield, and results of test cross, were selected for hybridization with two CMS lines; G46A and IR69625A. In the next year 4 iso-cytoplasmic restorer lines were grown; a commercial hybrid rice variety (Egyptian Hybrid 1), maintainer lines, and new hybrids were produced from it for an evaluation in a randomized complete block design (RCBD) with three replications compared to during 2020 (Figure 1).

Hybridization between parents (Line × Tester) was carried out following the technique proposed by [5]; ten hybrid combinations were generated through line × tester mating design of the two lines as female parents with 5 lines as pollinated parents in the 2019 rice-growing season.

The standard package of recommended practices was adopted for good crop growth. Five random plants from the central rows in each replication were selected and evaluated for yield and their component traits. Data were collected on pollen-fertility percentage (PF), spikelet-fertility percentage (SF), days to 50% heading (day) (HD), plant height (cm) (PH), panicle length (cm) (PL), number of panicles per plant (P/P), number of spikelets per panicle (Sp/P), number of filled grains per panicle (FG/P), panicle weight (g) (PW), 1000-grain weight (g) (1000-GW), grain yield per plant (g) (GY/P), anther length (mm) (AL), anther breadth (mm) (AB), filament length (mm) (FL), grain length (GL), grain width (GW), grain shape (GS), grain (kernel) length (KL), grain (kernel) width (KW), grain (kernel) shape (KS), grain elongation (GE), hulling percentage (H), milling percentage (M), and head rice-recovery percentage (HR). All the measurement techniques were based on the Standard Evaluation System of rice, International Rice Research Institute (IRRI) [12].

### 2.3. Statistical Analysis

#### 2.3.1. Estimation of Genetic Components

Analysis of variance (ANOVA) was used based on the model proposed by [13] to analyze the data statistically. The variance components were estimated from the analysis of variance as described by [14,15]. According to the formulas proposed by [16], the genetic, phenotypic, coefficient of genetic variance (GCV%), and coefficient of phenotypic variance (PCV%) were calculated. Broad-sense heritability a [h^2^_(bs)_] was calculated by the formula given by [17], as suggested by [18]. The genetic advance (GA) was estimated by using the heritability estimates, by the formula given by [18].

The GCV and PCV value were classified as follows: low = 0–10%; moderate = 10–20%; and high ≥ 20%, according to [19].

Broad-sense heritability was categorized as follows: low = 0–30%; medium = 30–60%; and high = above 60%, according to [18].

#### 2.3.2. Estimation of Genetic Parameters

Data were analyzed using analysis of variances of parental lines and hybrids for RCBD as suggested by [13] to test the significance of differences among the genotypes. Line × tester analysis was performed according to [20]. General combining ability (GCA) effects for each female or male parent and specific combining ability (SCA) effects for each cross combination were estimated according to [14]. The genetic components were estimated based on the expectations of mean squares according to [21].

#### 2.3.3. The Advantage over Commercial Variety

The increase or decrease in the newly developed restoration lines compared with the commercial variety (CK) was calculated as a percentage of the extent of distinction or superiority over the commercial variety, according to [22] and modified by [23].

To test the significance of the superiority or the advantage over commercial variety, values of L.S.D. were calculated according to the method suggested by [24] and modified by [23].

#### 2.3.4. Estimation of Heterosis

Heterosis was determined for each cross-over better-parent (BP), mid-parents (MP), and standard heterosis (SH) [22,25]. To test the significance of the heterosis effects for better-parent, mid-parents, and standard heterosis, values of L.S.D. were calculated according to the method suggested by [24] and modified by [23].

## 3. Results

### 3.1. Development of New Iso-Cytoplasmic Rice-Restorer Lines

#### 3.1.1. Mean Performance

The obtained results revealed that the iso-cytoplasmic restorer line NRL79 was the earliest, while it showed the lowest value for days to 50% heading (Appendix A). The iso-cytoplasmic restorer lines NRL80 and NRL81 gave the highest values of days to 50% heading, followed by NRL82. For grain yield, the new restorer lines NRL82, NRL81, NRL79, and NRL80 showed the highest values, respectively, compared to the check variety Gia178. Concerning spikelet-fertility restorer lines, NRL81, NRL80, NRL79, and NRL82 showed the highest spikelet fertility (Appendix A). The restorer lines showed the lowest desirable values compared to the check variety Gia178, respectively, for grain shape (Appendix A).

#### 3.1.2. Analysis of Variance of Promising Restorer Lines

The results presented in Appendix A of analysis of variance for promising restorer lines and check variety showed highly significant differences among the genotypes for all characters studied.

#### 3.1.3. Phenotypic (PCV%), Genotypic Coefficient of Variation (GCV%) and Genetic Advance

The traits P/P, Sp/P, FG/P, GY/P, AL, and FL showed high estimated values of phenotypic (PCV%) and genotypic coefficient of variation (GCV%). Moreover, the traits PH, PL, PW, AB, and GE recorded a moderate value of (PCV%) and (GCV%), while lower values of the (PCV%) and (GCV%) were observed for HD, PF%, SF%, 1000-GW, H%, M%, HR%, GL, GW, GS, KL, KW, and KS (Appendix A and Figure 2).

The characters PH, P/P, PL, Sp/P, FG/P, PW, GY/P, AL, AB, FL, and GE were obtained with high estimates of genetic advance as a percent of the mean (expected) (GA%). Moreover, moderate genetic advances were observed for 1000-GW, GW, GS, KW, and KS, while low genetic advances were observed for HD, PF%, SF%, H%, M%, HR%, GE, and KL (Appendix A and Figure 2).

The traits PH, P/P, PL, Sp/P, FG/P, PW, GY/P, AL, AB, FL, and GE showed high heritability with high (GA%). On the other hand, the traits HD, PF%, SF%, H%, M%, HR%, GL, and KL showed high heritability with low (GA%).

#### 3.1.4. The Advantage Percentage over Commercial Variety

The newly developed restorer lines showed to be significant and highly significant for all the studied characters except KS, of the percentage of advantage over Giza 178 commercial variety (commercial restorer) (Table 2). For pollen-fertility percentage and SF%, the lines NRL81 showed a highly significant positive percentage of advantage estimates over Giza 178. Concerning 1000-GW, all lines under this study showed highly significant positive estimate values over commercial restorer Giza 178; the percentage of advantage ranged from 17.3% to 31.2% for the NRL80 and NRL82, respectively. On the contrary, most of the studied lines recorded significant and highly significant positive estimates over Giza 178 for grain yield and its components; the values of percentage of advantage for grain yield ranged from 51% for the NRL80 to 100.4% for NRL82. The newly developed restorer lines showed significant and highly significant values of the percentage of advantage over Giza 178 for anther breadth, anther length, and filament length (Table 2). For anther length, the lines NRL79 and NRL82 showed significant and highly significant estimated values. Concerning anther breadth, the new restorer lines NRL79, NRL80, and NRL81 gave significant estimated values, while the line NRL82 gave highly significant estimated values. For filament length, the all-new developed restorer lines showed highly significant estimated values (Table 2).

Concerning the grain width of paddy (rough) rice, the promising lines showed highly significant positive estimated values over Giza 178, and the percentage of advantage ranged from 10.4% for the NRL81 to 18.4% for NRL82, respectively. Meanwhile, for the grain type (shape) of paddy (rough) rice, the three new promising lines showed highly significant negative values in comparison with Giza 178 as a commercial restorer; the percentage of advantage ranged from −7.7% for the NRL80 to −15.2% for NRL79, respectively, (Table 2).

### 3.2. Development of New Rice Hybrids

#### 3.2.1. Test Cross Experiment

The pollen fertility percentage of tested hybrids varied from 91.37% (G46A × Giza 178) to 98.22% (G46A × NRL80). On the contrary, the spikelet-fertility percentage of tested hybrids was varying from 87.35% (G46A × NRL82) to 95.08% (G46A × NRL80) (Appendix A).

#### 3.2.2. Evaluation of New Hybrids and Parental Lines

The data showed that IR69A × NRL80 and G46A × NRL81 showed the highest values of pollen fertility (%), respectively (Appendix A). Concerning, HD and PH, most of the new hybrids showed values acceptable and like the check hybrid. For P/P, five hybrids—IR69A × NRL82, G46A × NRL82, G46A × NRL80, IR69A × NRL79, and G46A × NRL81—showed the highest values, respectively. Regarding PL, the hybrids G46A × NRL80 and G46A × NRL81 showed the highest values, respectively. Concerning Sp/P, eight hybrids showed the highest values over hybrid commercial variety. The hybrids G46A × NRL79, G46A × NRL80, and G46A × Giza 178 showed the highest values of Sp/P, respectively (Appendix A and Figure 3). For FG/P, six hybrids showed the highest values over hybrid commercial variety. The hybrids G46A × NRL79, G46A × NRL80, and G46A × Giza 178 showed the highest values of FG/P, respectively (Appendix A and Figure 3). The highest values of PW were observed in seven hybrids: the hybrids G46A × NRL80, G46A × NRL79, and G46A × NRL81 (Appendix A and Figure 4). For 1000-GW, all hybrids showed values higher than the hybrid commercial variety. The highest values were observed in the hybrids IR69A × NRL82, IR69A × NRL81, G46A × NRL81, and G46A × NRL82, respectively (Appendix A and Figure 5). The data showed that all hybrids under study showed values higher than hybrid commercial variety for GY/P. The highest values were observed in the hybrids IR69A × NRL82, IR69A × NRL81, G46A × NRL79, IR69A × NRL79, G46A × NRL80, and G46A × NRL81, respectively, (Appendix A and Figure 6).

In the data obtained, eight hybrids under study showed values higher than the hybrid commercial variety for anther length. On the contrary, the data showed that all hybrids under study showed mean values higher than hybrid commercial variety for anther breadth and filament length (Appendix A).

The data showed that seven promising hybrids showed the lowest desirable values in comparison with hybrid commercial-variety grain length of paddy (rough) rice; the lowest values were observed for G46A × NRL80 and IR69A × NRL79, respectively, (Appendix A). Concerning grain width of paddy (rough) rice, the data showed that all promising hybrids showed the highest values over the hybrid commercial variety, with the values ranging from 2.90 for the IR69A × NRL79 to 3.38 for G46A × NRL81, respectively. On the other hand, regarding the grain shape (type) of paddy (rough) rice, the data showed that all promising hybrids showed the lowest desirable values in comparison with the hybrid commercial variety; the lowest values were observed of G46A × NRL81 and G46A × NRL80, respectively, (Appendix A). For means of kernel length, the data showed that all promising hybrids showed the lowest desirable values compared with the hybrid commercial variety; the lowest values were observed for G46A × NRL79 and G46A × NRL80, respectively (Appendix A). On the contrary, for the means of kernel width, the data showed that eight promising hybrids showed the highest mean values compared with the hybrid commercial variety; the highest mean values were observed for G46A × NRL82, G46A × Giza 178, and G46A × NRL79, respectively. Regarding kernel type, the data showed that all promising hybrids showed the lowest desirable values compared with the hybrid commercial variety; the lowest values were observed for G46A × NRL79, G46A × NRL81, G46A × NRL80, G46A × Giza 178, G46A × NRL82, and IR69A × NRL81, respectively. Concerning grain elongation, the data showed that four promising hybrids showed the highest values over the hybrid commercial variety, the highest mean values were observed of IR69A × NRL81, G46A × NRL80, IR69A × NRL79, and G46A × Giza 178, respectively. Regarding hulling percentage, the data showed that six promising hybrids showed the highest values over the hybrid commercial variety; the highest mean values were observed for IR69A × NRL81, G46A × NRL82, and G46A × NRL80, respectively. For the means of milling percentage, the data showed that five promising hybrids showed the highest values over the hybrid commercial variety; the highest mean values were observed of G46A × NRL80, G46A × NRL81, IR69A × NRL81, and G46A × NRL82, respectively. Regarding head rice-recovery percentage, the data showed that six promising hybrids showed the highest values over the hybrid commercial variety; the highest mean values were observed of G46A × Giza 178, G46A × NRL81, G46A × NRL80, and G46A × NRL82, respectively (Appendix A).

#### 3.2.3. Analysis of Variance for New hybrids and Parental Lines

All studied traits showed highly significant values of the mean squares of the genotypes and parents (Appendix A). Similarly, the mean squares of the parents vs. crosses showed significant and highly significant differences for studied traits, except for PF%, PH, PL, Sp/P, FG/P, GY/P, AL, AB, FL, GL, KS, and M%. The mean squares of the crosses (except for GL), lines (except for PF%, SF%, AL, AB, FL, GL, and GE), testers (except for PL, AL, GL, GS, KL, KS, and H%), and line × tester (except for PW, AL, AB, FL, GL, GW, GS, KL, KW, and KS) showed significant and highly significant differences (Appendix A).

The ratio of K^2^ GCA/K^2^ SCA was more than unity for grain yield; yield contributing traits and grain quality studied traits except for PL, PW, 1000—GW, GL, GW, GS, KL, KW, KS, and H%, were less than unity (Table 3).

#### 3.2.4. Estimation of Combining-Ability Effects

##### General Combining-Ability Effects

General combining-ability (GCA) effects for female lines (lines) is presented in Table 4; the female line G46 gave highly significant desirable values for HD, P/P, PL, Sp/P, FG/P, PW, 1000-GW, GY/P, GL, GW, GS, KL, KW, KS, H%, M%, and HR%, while the line IR69A gave highly significant desirable values for PF%, PH, and GE, and significant desirable values for AL (Table 4).

Moreover, the testers NRL79 gave the highest highly significant desirable values for HD, PH, PL, GL, KL, and KS (Table 5), while the testers NRL81 gave the highest highly significant desirable values for PF%, SF%, PL, PW, GY/P, GW, GS, KW, H%, and M%. The testers NRL82 gave the highest highly significant values for P/P, 1000-GW, AL, AB, and FL, while the testers Giza 178 gave the highest highly significant values for Sp/P, FG/P, and GE. On the other hand, the testers NRL68 gave the highest highly significant values for head rice percentage (Table 5).

##### Specific Combining Ability (SCA) Effects

From the data in Table 6, the hybrids IR69A × NRL80, G46A × NRL81, and G46A × NRL82 recorded a highly significant positive value of specific combining ability effects for PF percentage. Concerning SF percentage, the hybrids IR69A × NRL80 and G46A × NRL81 showed a highly significant positive value of specific combining-ability effects. For HD, two hybrid combinations—IR69A × Giza178 and G46A × NRL82—recorded highly significant negative (desirable) values of specific combining-ability effects (Table 6). Concerning PH, five hybrids showed significant and highly significant negative (desirable) values of SCA effects. The hybrid combinations G46A × NRL80, IR69A × NRL81, G46A × NRL79, and IR69A × Giza 178 showed the highest significant negative values. The data showed that four hybrid combinations—IR69A × NRL79, G46A × NRL81, G46A × NRL80, and IR69A × Giza 178—recorded highly significant positive and desirable values for P/P (Table 6). Two hybrid combinations—IR69A × NRL79 and G46A × NRL80—recorded highly significant positive values of SCA effects for PL. Five hybrid combinations—G46A × Giza 178, G46A × NRL80, IR69A × NRL79, IR69A × NRL82, and IR69A × NRL81—showed highly significant positive values of SCA effects for Sp/P, and FG/P (Table 6).

Concerning PW, two hybrids—G46A × NRL81 and IR69A × NRL82—recorded significant positive and desirable values of SCA effects. The data revealed that two hybrids—IR69A × NRL80 and G46A × NRL82—showed highly significant positive values of SCA effects, for 1000-GW. Concerning GY/P, five hybrids—G46A × NRL81, IR69A × NRL79, IR69A × NRL82, G46A × NRL80, and IR69A × Giza 178—gave highly significant and significant positive values of SCA effects (Table 6). The data showed that the studied hybrids did not show significant values for AL, AB, and FL, except the hybrid G46A × Giza 178, which gave significant value for anther breadth (Table 6). Concerning GL, the data showed that the cross combinations IR69A × NRL79 and G46A × NRL80 recorded significant negative desirable values of SCA effects (Table 6). Regarding GW, the cross combinations IR69A × NRL82 and G46A × Giza 178 showed highly significant and significant positive values of SCA effects. The data showed that the hybrids IR69A × NRL82 and G46A × Giza 178 recorded highly significant negative values of SCA effects for grain shape. Regarding GE, five hybrids—G46A × Giza 178, G46A × NRL80, IR69A × NRL81, IR69A × NRL79, and IR69A × NRL82—gave highly significant positive values of SCA effects. Concerning H%, the hybrids G46A × NRL80 and IR69A × NRL81 recorded highly significant and significant positive values of SCA effects. For M%, the hybrids G46A × NRL80, IR69A × NRL81, IR69A × Giza 178, G46A × NRL82, and IR69A × NRL79 recorded highly significant and significant positive values of SCA effects (Table 6).

The results in Table 7 revealed the proportional contribution of lines, testers, and line × tester interaction for the expression of traits. The results showed that lines played important role towards plant height (66.09%), panicle length (58.96%), spikelets/panicle (56.37%), filled grains/panicle (55.87%), panicle weight (47.60%), grain length (61.37%), grain types (72.31%), kernel width (62.71%), kernel types (67.11%), and head rice percentage (55.75%).

Meanwhile, the contribution of testers was more important for filament length (99.20), anther length (98.75), anther breadth (80.19), 1000-grain weight (79.58%), pollen-fertility percentage (61.50%), kernel length (60.89%), spikelet-fertility percentage (52.16%), milling percentage (52.98%), and grain yield/plant (45.48%). The contribution of maternal and paternal interaction (line × tester) was more important for panicles/plant (74.49%), grain elongation (52.00%), days to heading (47.88%), hulling percentage (39.39%), and grain width (37.52%), Table 7.

#### 3.2.5. Estimation of Heterosis Effects

The hybrid IR69A × NRL80 showed the highest highly significant positive values of the better-parent heterosis (BP), mid-parents heterosis (MP), and standard heterosis (SH) for PF% and SF%, (Appendix A). The studied hybrids showed positive values of BP heterosis, MP heterosis, and SH for HD. The BP heterosis did not show highly significant or significant negative values for PH. However, the hybrids IR69A × NRL82 and IR69A × NRL81 showed the highest highly significant and negative desirable values for MP and standard heterosis, respectively. The hybrids IR69A × NRL79, G46A × NRL80, and G46A × NRL81 showed the highest highly significant positive values in BP, MP, and SH heterosis for P/P, nonrespectively (Appendix A). Concerning PL, PW, 1000-GW and GY/P, GW, and GS, data revealed that the hybrid G46A × NRL81 showed the highest values for BP, MP, and SH heterosis, respectively. The hybrid G46A × Giza 178 showed the highest values of BP, MP, and SH heterosis for Sp/P, FG/P, and GE. Moreover, data revealed that the hybrids G46A × NRL82, G46A × NRL81, IR69A × NRL81, IR69A × NRL82, IR69A × NRL80, G46A × NRL80, and G46A × Giza 178 showed highly significant positive values of SH heterosis for 1000-GW, respectively (Appendix A). While the six hybrids showed values for GY of SH heterosis greater than 15% over the check-variety Egyptian hybrid one, these hybrids were G46A × NRL81 (125.1%), G46A × NRL80 (66.9%), IR69A × NRL79 (47.2%), G46A × NRL79 (24.6%), IR69A × NRL81 (23.4%), and IR69A × NRL82 (16.2%). Concerning AL, AB, and FL the hybrids IR69A × NRL82 and G46A × NRL82 showed the highest highly significant values for SH heterosis. The hybrids G46A × NRL81, G46A × NRL80, G46A × NRL79, and G46A × NRL82 showed the highest desirable values of SH heterosis for GL, GW, GS, KL, KW, and KS. Obviously, H% of the hybrid G46A × NRL80 showed highly significant and significant positive values for SH and MP heterosis, respectively. The hybrid G46A × NRL82 showed significant and highly significant positive values BP, MP, and SH heterosis for M% and HR% (Appendix A).

## 4. Discussion

### 4.1. Development of New Iso-Cytoplasmic Rice-Restorer Lines

#### 4.1.1. Mean Performance

Awad-Allah (2011) [26] identified the parental lines Giza 178 and BG 34-8 as restorer lines that have the band of M2 as a dominant marker linked with an allele of the *Rf1* gene on chromosome 1. Moreover, [26] found a band detected by an RM 171 marker in the parental lines Giza 178 and BG 34-8. These results suggest that these lines may have the allele of the *Rf4* gene, which is known to be linked with RM 171 marker on chromosome 10 in WA CMS lines. Awad-Allah (2011) [23,26] selected the promising hybrids IR69A × Giza178 and G46A × BG 34-8 to grow to produce F_2_, and the selection started in F_2_ up to F_7_, and the new selections were grown along with the parental lines and evaluated for phenotypic performance and yielding ability. The newly developed restorer lines contain restorer genes from the parents.

The highest proportion of iso-cytoplasmic restorer lines are among the Egyptian Hybrid 1 (IR69A × Giza 178) (3 lines), followed by a promising rice hybrid (G46A × BG34-8) (1 line). The iso-cytoplasmic restorer lines derived from IR69A × Giza 178 were the earliest (102 days). For the restorer line to be effective and desirable, restorer lines should have a synchronized flowering period with cytoplasmic male sterile lines. Therefore, lines with ~100 days to 50% heading are most desirable. In the restoration lines, the plant height should be more than the cytoplasmic male sterile lines. Because the PH of popular and promising cytoplasmic male sterile lines IR69A, G46A, IR79156A, and IR 58025A are less than and around 100 cm approximately, in this case, the plant height of restorer lines must be around 110–125 cm at least; similar results were obtained by [23].

The most important characteristic for comparing the performance of restoration lines is the characteristic of the grain yield, because it reflects the performance of all the attributes of the components. The new restorer lines derived from the two hybrids were found to possess a higher overall mean yield. The new iso-cytoplasmic restorer lines under this study are derived from rice hybrids through selfing pollinated and continuous selection, and therefore they contain sterile cytoplasm from the female line (CMS). The lines carrying WA cytoplasm have been observed to show incomplete panicle excretion because wild abortive cytoplasm (WA) has a significant effect on panicle excretion [23]. This explains the existence of variation in the extent of panicle excretion in each generation, but there is an opportunity to correct the improvement of panicle excretion in restoration lines by applying appropriate selection pressure at the level of panicle excretion in segregating generations [10]. The second most important characteristic that has a strong effect on the yield is the spikelet fertility, which helps in producing a better yield. Restorer lines NRL81, NRL80, and NRL79 produced (derived) from IR69A × Giza 178 and NRL82 produced (derived) from G46A × BG34-8 have shown the highest spikelet fertility.

Based on their performances and phenotypes, they were selected as well as evaluated as iso-cytoplasmic restorers. The identification of lines can be screened for the presence of fertility genes and their fertility-restoration behavior by crossing with CMS lines [7]. In addition, based on the association among different traits observed, it can be realized that simultaneous selection for P/P, PL, and SF would enable the improvement of GY/P [10]. Thus, the selection of P/P followed by PL and 1000-GW would help in increasing the yield as they were both reciprocally and directly correlated with the grain yield. Traits that show higher variability can provide higher genetic gains in breeding programs and have been used in rice-breeding programs to split the observed variation and study the relationships between different traits, [27]. This method has been instrumental in developing new iso-cytoplasmic restorer lines. The promising iso-cytoplasmic restorer lines assist in the development of heterotic hybrids, and the basic group (core set) of iso-cytoplasmic restorers can be used for additional improvement in the restorer at the same time [10].

#### 4.1.2. Analysis of Variance of Promising Restorer Lines

The genotypes studied showed highly significant differences for all studied traits in ANOVA analysis Appendix A, suggesting that every genotype is genetically divergent and there is ample scope for selection of characters from these diverse sources for studied traits; this shows that there is variability between the studied lines and genotypes as well as a positive response to the selection. The presence of genetic variability is a prime requirement in the rice-improvement program. The set of genotypes used in the present study indicate the existence of significant differences among themselves for all the studied traits; these findings follow the findings of [28,29,30,31,32,33,34].

#### 4.1.3. Phenotypic Coefficient of Variation (PCA%), Genotypic Coefficient of Variation (GCA%), and Genetic Advance

The results in Appendix A revealed the existence of a considerable amount of variability in all the studied characters among the genotypes. In this study, the PCA% values were higher than the GCA% values, which indicate that there is an influence of the environment on the expression of these traits. However, there was little difference between the values of the phenotype coefficients (PCA%) and the values of the genotype coefficients (GCA%) for variance in all the studied traits, and this indicates that there is a limited role for environmental variance in the expression of these characters, Appendix A and Figure 2. It is known that genetic variability is a condition for selecting genotypes that are superior to the existing cultivars. Therefore, selection based on the genotypic performance of the traits would be effective to bring about considerable improvement in these characters. In the breeding programs, the selection is based on measurements of phenotypic character, and genotypic variability is measured through analysis of variance; similar findings were observed by [23,28,33,35].

A high estimate of PCA% and GCA% for GY/P and five studied traits, moderate for five characters, and lower for 13 traits were observed; this finding is expected due to the concentration of breeder selection for selection to a limited class, which leads to less variation (Appendix A and Figure 2); similar results were obtained by [23,28].

Johnson et al. (1955) [18] reported that genetic advance is a useful indicator of the progress that can be expected because of selection on the related population, while heritability in conjunction with genetic advance would give a more reliable index of selection value. High heritability with high genetic advance as percent of the mean (expected) was observed for 11 characters; this indicates that heritability is most likely due to additive gene-action effects and selection may be effective. These results indicate that there is a lot of genetic improvement in the lines for these traits for further selection and subsequent use in the breeding program.

Furthermore, high heritability with moderate genetic advances was observed for five characters (Appendix A). Moreover, high heritability with low genetic advance was observed for eight traits, which indicates the presence of nonadditive gene action for the expressions of these traits. High heritability has been observed but with a high influence of environment rather than genotype, and selection for such traits may not be rewarding. Similar results were also reported by [23,28,29,30,31,32,33,35,36,37].

#### 4.1.4. The Advantage over Commercial Variety

The significant and highly significant values of percentage of advantage over Giza 178 commercial variety (the only commercial restorer) were observed among the genotypes for all the studied traits, demonstrating that the selection is efficient in the genetic improvement for these traits (Table 2), such as 1000-grain weight with ranges from 17.3% for the NRL80 to 31.2% for NRL82. The use of these lines may be useful as restoration lines to produce new, promising, and desirable hybrid rice varieties for farmers in Egypt. This is the main defect of a commercial hybrid, i.e., Egyptian Hybrid 1 (IR69A × Giza 178). While the new restoration fertility lines showed significant and highly significant positive estimates higher than Giza 178 for GY/P, AL, AB, and FL, this improvement in these traits will lead to an increase in outcrossing between the parental lines in seed production and thus increase the yield (quantity) of the produced seeds. These lines can be used as inbred cultivars or inbred varieties and restoration fertility lines to develop promising new hybrid rice varieties in Egypt. These findings agreed with results reported by [10,23,28,38,39,40,41].

### 4.2. Development of New Rice Hybrids

#### 4.2.1. Experiment of Test Cross

A test cross-trial was evaluated to identify restorer fertility lines and maintainer lines; this trial is the first step in a hybrid rice-breeding program to develop new hybrids. Pollen fertility and SF analysis are used to identify restorer fertility and maintainer lines [42]. First, a test cross (data not presented) was conducted to select the best restorer lines that gave high restoration ability and produce hybrids with high PF% and SF%. The highest four lines in the restoration ability were selected and the crossing was conducted to study and evaluate the new hybrids for floral traits, grain quality traits, grain yield, and its components. Based on the results of the test cross trial, the four promising lines were identified as effective restorer fertility lines for two CMS lines, (Appendix A). It will be useful to use highly fertile hybrids in developing and releasing new and promising hybrids [10,26,42,43,44,45,46].

#### 4.2.2. Evaluation of New Hybrids and Parental Lines

The newly developed hybrids under this study showed values higher than the hybrid commercial variety of 1000-grain weight (g), grain yield/plant (g), grain width of paddy (rough) rice, and grain shape (type) of paddy rice (Appendix A, Figure 5 and Figure 6). This is useful in using these promising hybrids in released new hybrids and is used in developing new promising restorer lines; similar results were obtained by [23,26,42,43,44,45,46,47].

#### 4.2.3. Analysis of Variance for New Hybrids and Parental Lines

The mean squares of the genotypes and parents for studied traits showed highly significant values in ANOVA analysis (Appendix A). Based on this result, there are significant differences between the genotypes. Equally, the mean squares of the parents vs. cross, crosses, lines, tester, and line × tester showed significant and highly significant differences, except for some of the studied traits; parents vs. crosses mean square is an indication of overall average heterosis crosses. Similar results were obtained by [38,48,49,50,51,52,53].

The ratio of K^2^ GCA/K^2^ SCA was more than unity for contributing traits, grain yield, and grain quality (Table 3), indicating a preponderance of additive-gene effects in the expression of these traits, while the ratio of K^2^ GCA/K^2^ SCA was less than unity, indicating preponderance of non-additive gene effects in the expression in these crosses of PL, PW, 1000–GW, AL, AB, FL, GL, GW, GS, KL, KW, KS, and H%. Then, selection procedures based on the accumulation of additive effects would be effective in improving these traits. These findings agreed with those obtained by [33,44,53,54].

#### 4.2.4. Estimation of Combining Ability Effects

##### General Combining Ability Effects

Evaluation of (GCA) provides a tool selection for crop breeders to select good parental lines for hybridization. Moreover, it is a powerful method to clarify the nature of gene action for preferred characters [20]. The results of this study showed that the parental lines were identified as the best combiner for at least one of the studied traits and a good combiner for a minimum of two yield-related characteristics (Table 4 and Table 5). Among these, the female line G46A (CMS) gave highly significant desirable values of (GCA) for HD, P/P, PL, Sp/P, FG/P, PW, 1000-GW, GY/P, GL, GW, GS, KL, KW, KS, H%, M%, and HR%, while the female line IR69A (CMS) gave highly significant desirable values for PF%, PH, GE, and significant desirable values for AB (Table 4). This means that these lines are good general combiners for these traits. Similar results were obtained by [10,26,43,44,45,46,47,53,55].

On the contrary, the testers NRL79 gave the highest highly significant desirable values of GCA for HD, PH, PL, GL, LL, and KS (Table 5), while the tester NRL81 gave the highest highly significant desirable values for PF%, SF%, PL, PW, GY/P, GW, GS, KW, H%, and M%. On the contrary, the tester NRL82 gave the highest highly significant desirable values for P/P, 1000-GW, AL, AB, and FL, while the tester Giza 178 gave the highest highly significant desirable values for Sp/P, FG/P, and GE. Moreover, the tester NRL80 showed the highest highly significant desirable values for HR% (Table 5). Thus, a good GCA estimate could help in identifying the lines and testers that would give hybrids and improve parental lines for desirable traits. These results agreed with those obtained by [10,26,42,44,45,46,53,55,56].

##### Specific Combining Ability (SCA) Effects

All the hybrids studied showed significant and highly significant positive SCA effects for at least one yield-related trait (Table 6). The desirable highest significant value of SCA effects has been shown by hybrid IR69A × NRL80 for PF%, SF%, and 1000-GW, while the hybrid IR69A × Giza 178 for HD, as well as the hybrid IR69A × NRL79 for P/P, PL, the hybrid G46A × NRL80 for PH, H%, M%, the hybrid IR69A × NRL82 for 1000-GW, GW, GS, the hybrid G46A × Giza 178 for Sp/P, FG/P, AB, GE, and the hybrid G46A × NRL81 PW, GY/P (Table 6). The positive values indicated that the non-additive effects could be present in these hybrid combinations for studied traits (Table 6). The hybrid combinations appeared to be good combiners to improve restorer lines, rice cultivars, and hybrids for floral traits, grain quality, grain yield/plant, and its components; these findings agreed with other results obtained by [26,42,43,44,53,57].

The data showed that no hybrid combinations had positive SCA effects for all the studied traits (Table 6). This finding agreed with those reported in earlier studies [38,56,58]. Present results revealed that the two hybrid combinations showed a high significant SCA effect for GY and different traits had both parents with a high GCA effect. Such results showed the role of the cumulative effects of additive × additive interactions of positive alleles [26,43,56,57]. On the contrary, other hybrids revealed that significantly high SCA effects in desirable traits had at least one of the parents reflecting poor GCA effects. This may be due to a good combiner parent displaying suitable additive effects and a poor combiner parent producing epistatic effects [42,57,58,59,60,61]. Concurrently, good-by-good general combiners did not always present the best hybrids in terms of SCA. High SCA effects of the hybrids showing involving low/low general combiners indicate that the non-additive genetic effects and these hybrid combinations could be exploited for heterosis breeding programs [53]. It is concluded from the present results that there is the possibility to breed good hybrids of rice and rice cultivars with desirable traits and high yielding lines than the existing lines either through heterosis breeding or through recombinant breeding with selection in later generations to develop traits adaptable to high yielding parental lines of hybrid.

#### 4.2.5. Estimation of Heterosis Effects

The newly developed rice hybrids showed significant and highly significant desirable values in the BP heterosis, MP heterosis, and standard heterosis at least for yield components or desirable traits (Appendix A). Among these, the hybrid IR69A × NRL80 showed the highest positive values of PF% and SF% of the BP heterosis, MP heterosis, and SH. Pollen fertility and SF% are very important traits that directly affect yield in rice varieties and hybrids. Similar results were reported by many researchers among them [26,43,45,46,47,52,53,58].

Concerning PH, the hybrids that showed negative values may be good rice hybrids or may be useful to breed good rice cultivars. In contrast, the hybrids showed significant and highly significant positive values of BP heterosis, MP heterosis, and SH, which may be useful to breed good restorer lines. Similar results were found by many authors among them [26,46,51,52,58,61,62,63,64,65,66,67,68].

Furthermore, three, four, and eight hybrids showed highly significant positive values in BP heterosis, MP heterosis, and SH, for 1000-GW, respectively. In addition, 1000-GW is one of the most important traits that directly affect the potential for grain yield in rice varieties and hybrids. These results are similar to the results obtained by [26,43,45,46,47,51,52,58,61,62,63,64,65,66,67,68].

For GY/P three, four, and six hybrids showed highly significant and significant positive values in BP heterosis, MP heterosis, and SH, respectively. The highest value was detected in hybrids G46A × NRL81 and G46A × NRL80 for BP heterosis, MP heterosis, and SH, respectively. Moreover, the six hybrids showed values for SH of more than 15% over the Egyptian hybrid one as the check variety; these hybrids were G46A × NRL81, G46A × NRL80, IR69A × NRL79, G46A × NRL79, IR69A × NRL81, and IR69A × NRL82. Thus, these hybrids can be used for commercial use (Appendix A). In the previous studies, it was reported that the hybrids with high grain yield showed a high heterosis percentage [26,43,45,46,49,51,52,58,59,60,61].

Concerning grain width (breadth), data revealed that one, six, and nine hybrids showed highly significant and significant positive values in BP heterosis, MP heterosis, and SH, respectively. The highest values were shown in the hybrid G46A × NRL81 for BP heterosis, MP heterosis, and SH, respectively. Moreover, all studied hybrids showed values for SH heterosis over the check-variety Egyptian hybrid rice one, for this trait. Similar results were reported by [49,58]. Meanwhile, two, seven, and nine hybrids showed highly significant and significant negative desirable values in BP heterosis, MP heterosis, and SH for the grain type (shape), respectively. The hybrid G46A × NRL81 showed the highest values for BP, MP, and SH heterosis. Moreover, all studied hybrids showed desirable values for SH heterosis better than the check-variety Egyptian hybrid rice one for this trait (Appendix A). Similar results of this trait and grain quality traits were reported by [26,44,46,49,50,51,52,53,58].

## 5. Conclusions

The newly developed restorer fertility lines showed significant and highly significant values of percentage of advantage over the check (control) variety for all the studied characters, indicating that the selection is effective in the improvement of these traits. New restorer lines showed highly significant positive values over commercial restorer lines for grain yield, with values ranging from 51% for NRL80 to 100.4% for NRL82, respectively. Furthermore, three newly developed restorer lines showed highly significant negative desirable values of advantage over the check variety (Giza 178) for grain shape of paddy rice, with the values ranging from −7.7% for the NRL80 to −15.2% for NRL79, respectively. These lines could be used as restorer fertility lines to breed new promising hybrids and inbred rice varieties. Moreover, the six rice hybrids showed values for SH heterosis of grain yield/plant of more than 15% over the check hybrid variety with high values of 1000-grain weight and desirable grain shape; these hybrids were G46A × NRL81 (125.1%), G46A × NRL80 (66.9%), IR69A × NRL79 (47.2%), G46A × NRL79 (24.6%), IR69A × NRL81 (23.4%) and IR69A × NRL82 (16.2%).

## Figures and Tables

**Figure 1 genes-13-00808-f001:**
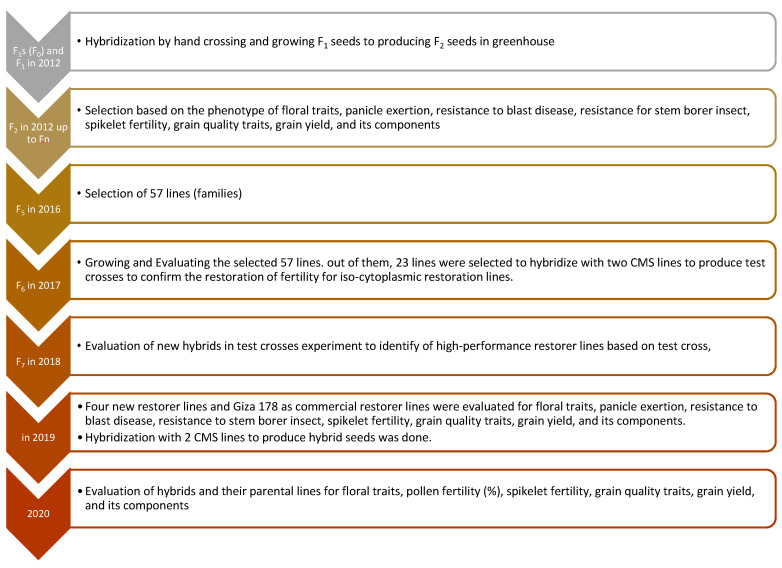
Breeding scheme illustrating the development of 4 promising new iso-cytoplasmic restorer lines in rice through Rice Breeding Program, Rice Research Section, Field Crops Research Institute, Agricultural Research Center, Giza, Egypt.

**Figure 2 genes-13-00808-f002:**
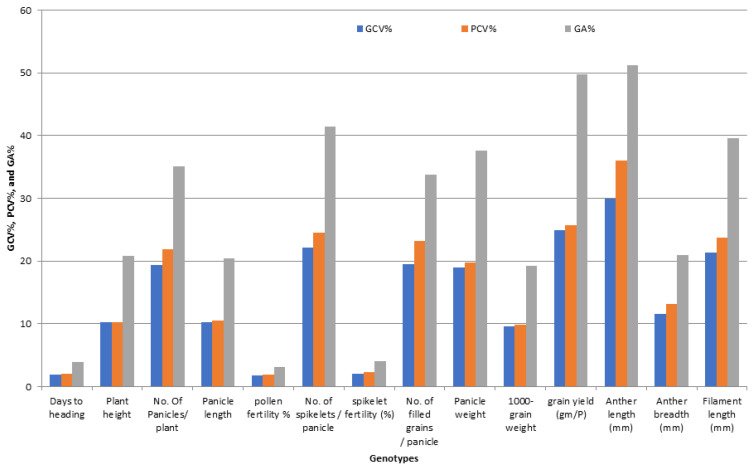
The percentage of variability parameters for floral traits, grain yield, and their contributing traits in promising restorer lines of rice.

**Figure 3 genes-13-00808-f003:**
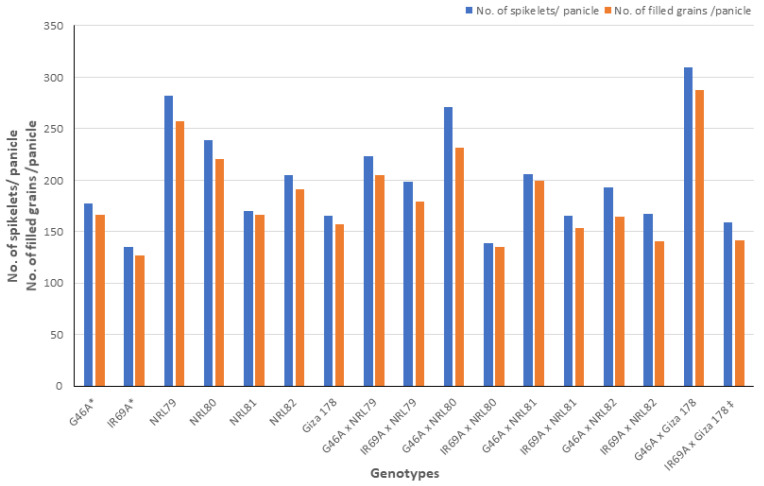
The mean performance for the number of spikelets per panicle and number of filled grains per panicle of the studied genotypes. *: The traits were recorded on maintainer lines that were related to CMS lines, ‡: Hybrid check.

**Figure 4 genes-13-00808-f004:**
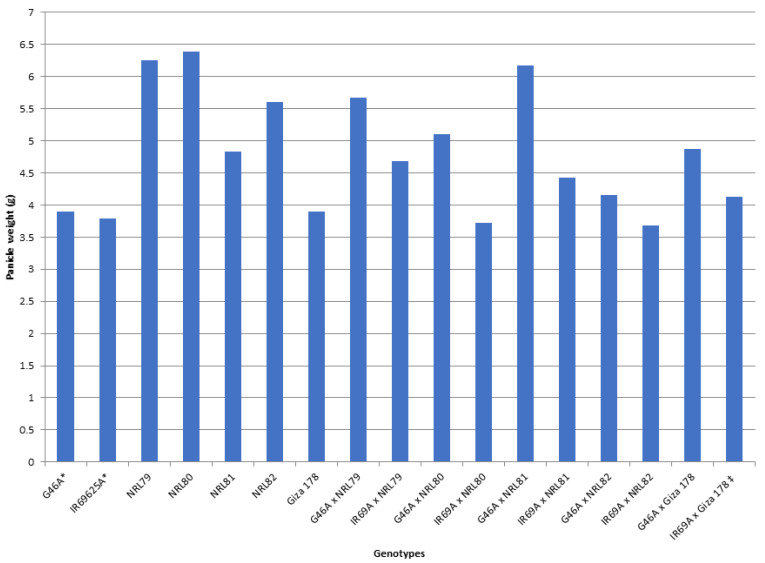
The mean performance for the panicle weight of the studied genotypes. *: The traits were recorded on maintainer lines that were related to CMS lines, ‡: Hybrid check.

**Figure 5 genes-13-00808-f005:**
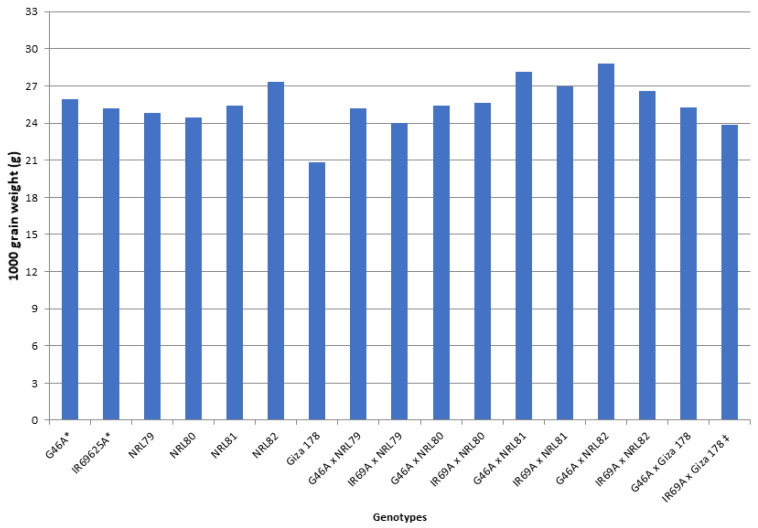
The mean performance for the 1000-grain weight of the studied genotypes. *: The traits were recorded on maintainer lines that were related to CMS lines, ‡: Hybrid check.

**Figure 6 genes-13-00808-f006:**
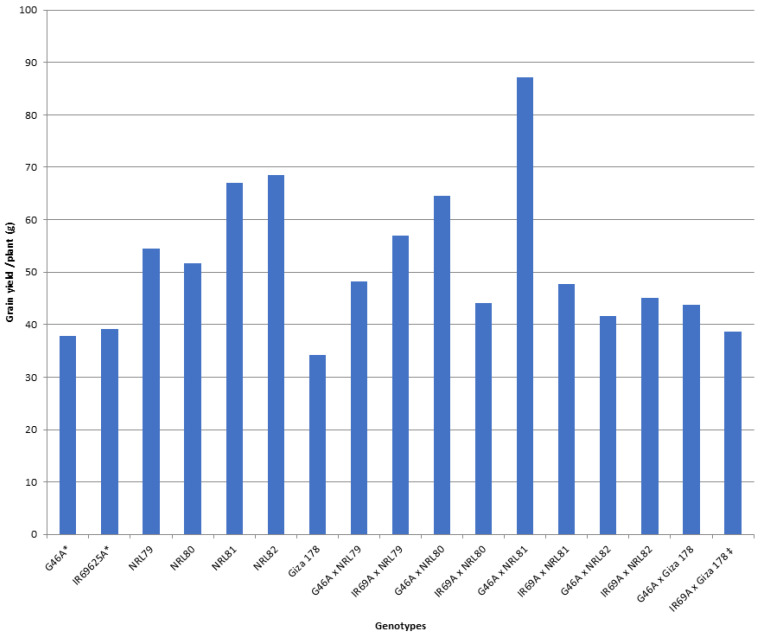
The mean performance for the grain yield/plant of the studied genotypes. *: The traits were recorded on maintainer lines that were related to CMS lines, ‡: Hybrid check.

**Table 1 genes-13-00808-t001:** Names and their parentage of the genotypes studied.

Name	Parentage
Gang46A (G46A)	Erjiu’ai 7/V41B//Zhenshan 97/Ya’aizao
Gang46B (G46B)	Erjiu’ai 7/V41B//Zhenshan 97/Ya’aizao
IR69625A (IR69A)	
IR69625B (IR69B)	
NRL 79	IR69A/Giza178
NRL 80	IR69A/Giza178
NRL 81	IR69A/Giza178
NRL 82	G46A/BG34-8
Giza 178 (local check)	Giza175/Milyang 49

**Table 2 genes-13-00808-t002:** Percentage of advantage over commercial variety for the grain yield, yield contributing traits, floral traits, and grain quality traits of promising restorer lines of rice.

Crosses	NRL79	NRL80	NRL81	NRL82	L.S.D. 5%	L.S.D. 1%
Traits						
Pollen fertility (%)	1.1 ^ns^	1.0 ^ns^	4.6 **	0.5 ^ns^	1.7	2.4
Spikelet fertility (%)	0.7 ^ns^	1.2 ^ns^	5.5 **	0.7 ^ns^	1.6	2.3
Days to heading	0.1 ^ns^	2.1 **	3.0 **	4.7 **	1	1.4
Plant height (cm)	7.9 **	14.9 **	15.2 **	31.5 **	2.5	3.6
No. of panicles/plant	4.5 ^ns^	12.9 ^ns^	45.2 **	55.1 **	3.2	4.7
Panicle length (cm)	12.2 **	13.1 **	−6.4 *	21.4 **	1.1	1.7
No. of spikelets/panicle	70.2 **	44.1 **	2.9 ^ns^	23.7 ^ns^	41.4	60.3
No. of filled grains/panicle	64.1 **	40.8 *	5.9 ^ns^	22.0 ^ns^	46.7	68
Panicle weight (g)	60.1 **	63.6 **	23.7 **	43.7 **	0.6	0.8
1000 grain weight (g)	19.2 **	17.3 **	22.1 **	31.2 **	1	1.4
Grain yield/plant (g)	59.4 **	51.0 **	95.9 **	100.4 **	6.5	9.5
Anther length (mm)	46.8 *	13.8 ^ns^	6.1 ^ns^	66.1 **	0.8	1.2
Anther breadth (mm)	20.0 *	15.0 *	17.5 *	40.0 **	0.06	0.08
Filament length (mm)	79.0 **	75.0 **	70.6 **	91.6 **	1.3	1.8
Grain length	−5.0 *	5.2 **	6.6 **	7.5 **	0.3	0.4
Grain width	12.3 **	14.1 **	10.4 **	18.4 **	0.2	0.3
Grain types	−15.2 **	−7.7 **	−3.5 ^ns^	−9.2 **	0.1	0.2
Kernel length	2.3 ^ns^	0.3 ^ns^	6.4 **	1.7 ^ns^	0.1	0.2
Kernel width	15.4 **	5.6 ^ns^	10.5 *	19.6 **	0.2	0.3
Grain elongation	21.2 **	23.1 **	55.1 **	49.0 **	6.1	8.8
Kernel types	−10.5 **	−4.9 ^ns^	−3.7 ^ns^	−15.0 **	0.1	0.2
Hulling (%)	−2.4 *	0.1 ^ns^	1.4 ^ns^	−0.4 ^ns^	1.3	2
Milling (%)	−0.6 ^ns^	0.5 ^ns^	2.9 **	−0.2 ^ns^	1	1.5
Head rice (%)	−7.2 **	-1.8 ^ns^	−0.6 ^ns^	−2.4 ^ns^	2	2.9

^ns^: Not significant, *: Significant, **: Highly significant.

**Table 3 genes-13-00808-t003:** The ratio between K^2^ GCA and K^2^ SCA for the grain yield, yield contributing traits, floral traits, and grain quality traits for the studied genotypes.

Genetic Components Traits	K^2^ GCA	K^2^ SCA	K^2^ GCA/K^2^ SCA
Pollen fertility (%)	0.6	4.5	4.5
Spikelet fertility (%)	−0.1	3.8	6.9
Days to heading	1.3	0.5	2.5
Plant height (cm)	83.2	12.4	26.6
No. of panicles/plant	8.3	7	73.3
Panicle length (cm)	1.5	0.2	0.6
No. of spikelets/panicle	2778.4	388.1	1816.1
No. of filled grains/panicle	2141.3	425.4	1146.5
Panicle weight (g)	0.6	0.3	0.05
1000 grain weight (g)	0.7	2.3	0.2
Grain yield/plant (g)	55.2	107	187.4
Anther length	−0.0050	0.0700	−0.0300
Anther breadth	0.0002	0.0015	0.0001
Filament length	0.0003	0.8420	−0.0532
Grain length	0.0100	0.0020	0.0080
Grain width	0.033	0.01	0.003
Grain types	0.033	0.004	0.003
Kernel length	0.01	0.01	0.0003
Kernel width	0.02	0.01	0
Kernel types	0.02	0.004	0
Grain elongation	1.1	82.1	179.9
Hulling (%)	0.5	0.3	0.7
Milling (%)	2.3	4	4.1
Head rice (%)	39.8	5.1	27.2

**Table 4 genes-13-00808-t004:** General combining ability effects of the lines for the grain yield contributing traits, floral traits, and grain quality traits.

Lines Traits	G46A	IR69A	L.S.D. 5%	L.S.D. 1%
Pollen fertility (%)	−0.61 **	0.61 **	0.35	0.49
Spikelet fertility (%)	−0.19 ^ns^	0.19 ^ns^	0.38	0.54
Days to heading	−0.83 **	0.83 **	0.15	0.22
Plant height (cm)	6.46 **	−6.46 **	0.47	0.66
No. of panicles/plant	2.05 **	−2.05 **	0.4	0.56
Panicle length (cm)	0.86 **	−0.86 **	0.14	0.2
No. of spikelets/panicle	37.3 **	−37.3 **	3.3	4.6
No. of filled grains/panicle	32.8 **	−32.8 **	3.6	5.1
Panicle weight (g)	0.5 **	−0.5 **	0.1	0.2
1000 grain weight (g)	0.6 **	−0.6 **	0.2	0.2
Grain yield/plant (g)	5.3 **	−5.3 **	0.9	1.2
Anther length	−0.02 ^ns^	0.02 ^ns^	0.07	0.11
Anther breadth	−0.01 *	0.01 *	0.01	0.01
Filament length	0.07 ^ns^	−0.07 ^ns^	0.1	0.15
Grain length	−0.08 **	0.08 **	0.04	0.06
Grain width	0.1 **	−0.1 **	0.02	0.03
Grain types	−0.13 **	0.13 **	0.02	0.03
Kernel length	−0.06 **	0.06 **	0.03	0.04
Kernel width	0.10 **	−0.10 **	0.02	0.03
Kernel types	−0.10 **	0.10 **	0.02	0.03
Grain elongation	−0.8 **	0.8 **	0.6	0.8
Hulling (%)	0.5 **	−0.5 **	0.2	0.3
Milling (%)	1.1 **	−1.1 **	0.3	0.4
Head rice (%)	4.5 **	−4.5 **	0.5	0.7

**: Highly significant at 1% *: Significant at 5% ^ns^: Nonsignificant.

**Table 5 genes-13-00808-t005:** General combining-ability effects of the tester lines for the grain yield, yield contributing traits, floral traits, and grain quality traits.

Tester Traits	NRL79	NRL80	NRL81	NRL82	Giza 178	L.S.D. 5%	L.S.D. 1%
Pollen fertility (%)	−0.2 ^ns^	0.8 **	3.3 **	−2.1 **	−1.9 **	0.5	0.8
Spikelet fertility (%)	0.03 ^ns^	1.2 **	2.7 **	−2.6 **	−1.3 **	0.6	0.9
Days to heading	−0.87 **	0.37 **	0.43 **	0.78 **	−0.72 **	0.24	0.34
Plant height (cm)	−4.6 **	−2.6 **	2.7 **	0.4 ^ns^	4.1 **	0.7	1
No. of panicles/plant	1.4 **	-0.6 ^ns^	0.5 ^ns^	3.0 **	−4.3 **	0.6	0.9
Panicle length (cm)	0.6 **	−0.2 *	0.6 **	−0.4 **	−0.5 **	0.2	0.3
No. of spikelets/panicle	8.3 **	2.5 ^ns^	-16.7 **	−22.4 **	28.3 **	5.2	7.3
No. of filled grains/anicle	9.7 **	−0.2 ^ns^	-5.7 *	−31.0 **	27.2 **	5.7	8.1
Panicle weight (g)	0.5 **	−0.3 *	0.6 **	−0.7 **	−0.2 ^ns^	0.2	0.3
1000 grain weight (g)	−1.4 **	−0.5 **	1.6 **	1.7 **	−1.4 **	0.2	0.3
Grain yield/plant (g)	0.8 ^ns^	2.6 **	15.7 **	−8.5 **	−10.5 **	1.4	1.9
Anther length	0.23 **	−0.09 ^ns^	−0.27 **	0.39 **	−0.26 **	0.12	0.17
Anther breadth	0.01 ^ns^	−0.003 ^ns^	−0.01 ^ns^	0.06 **	−0.05 **	0.01	0.02
Filament length	0.26 **	0.23 *	0.28 **	0.83 **	−1.61 **	0.16	0.23
Grain length	−0.11 **	−0.05 ^ns^	0.11 **	0.04 ^ns^	0.01 ^ns^	0.07	0.1
Grain width	−0.04 **	−0.01 ^ns^	0.15 **	0.04 *	−0.14 **	0.03	0.04
Grain types	−0.003 ^ns^	−0.012 ^ns^	−0.080 **	−0.025 ^ns^	0.120 **	0.037	0.052
Kernel length	−0.11 **	−0.10 **	0.04 *	0.07 **	0.10 **	0.04	0.06
Kernel width	0.05 **	−0.08 **	0.11 **	-0.03 ^ns^	-0.05 **	0.03	0.04
Kernel types	−0.08 **	0.03 ^ns^	−0.07 **	0.04 **	0.08 **	0.03	0.04
Grain elongation	1.2 *	−0.6 ^ns^	−3.6 **	−11.0 **	14.0 **	0.9	1.3
Hulling (%)	−1.08 **	0.12 ^ns^	0.49 *	0.46 *	0.01 ^ns^	0.38	0.53
Milling (%)	−1.8 **	−1.2 **	3.0 **	1.3 **	−1.2 **	0.4	0.6
Head rice (%)	−3.4 **	2.7 **	0.8 *	1.3 **	−1.3 **	0.8	1.1

**: Highly significant at 1% *: Significant at 5% ^ns^: Nonsignificant.

**Table 6 genes-13-00808-t006:** Specific combining ability for the grain yield, yield contributing traits, floral traits, and grain quality traits of the crosses.

Crosses Traits	G46A × NRL79	IR69A × NRL79	G46A × NRL80	IR69A × NRL80	G46A× NRL81	IR69A × NRL81	G46A × NRL82	IR69A × NRL82	G46A × Giza 178	IR69A × Giza 178	L.S.D. 5%	L.S.D. 1%
PF (%)	0.62 ^ns^	−0.62 ^ns^	−2.64 **	2.64 **	1.21 **	−1.21 **	1.11 **	−1.11 **	−0.30 ^ns^	0.30 ^ns^	0.8	1.1
SF (%)	0.53 ^ns^	−0.53 ^ns^	−3.22 **	3.22 **	2.01 **	−2.01 **	0.75 ^ns^	−0.75 ^ns^	−0.06 ^ns^	0.06 ^ns^	0.9	1.2
DH	0.14 ^ns^	−0.14 ^ns^	−0.12 ^ns^	0.12 ^ns^	−0.17 ^ns^	0.17 ^ns^	−1.52 **	1.52 **	1.67 **	−1.67 **	0.3	0.5
PH	−3.29 **	3.29 **	−4.46 **	4.46 **	4.29 **	−4.29 **	1.38 *	−1.38 *	2.08 **	−2.08 **	1	1.5
P/P	−8.34 **	8.34 **	5.31 **	−5.31 **	6.00 **	−6.00 **	0.76 ^ns^	−0.76 ^ns^	−3.74 **	3.74 **	0.9	1.3
PL	−0.9 **	0.9 **	0.8 **	−0.8 **	0.2 ^ns^	−0.2 ^ns^	0.1 ^ns^	−0.1 ^ns^	−0.2 ^ns^	0.2 ^ns^	0.3	0.5
Sp/P	−25.0 **	25.0 **	28.9 **	−28.9 **	-16.8 **	16.8 **	−24.4 **	24.4 **	37.3 **	−37.3 **	7.3	10.3
FG/P	−19.6 **	19.6 **	17.6 **	−17.6 **	−11.7 **	11.7 **	−20.5 **	20.5 **	34.2 **	−34.2 **	8.1	11.4
PW (g)	−0.04 ^ns^	0.04 ^ns^	0.15 ^ns^	−0.15 ^ns^	0.35 *	−0.35 *	−0.30 *	0.30 *	−0.16 ^ns^	0.16 ^ns^	0.3	0.4
1000-GW	0.001 ^ns^	−0.001 ^ns^	−0.67 **	0.67 **	0.04 ^ns^	−0.04 ^ns^	0.51 **	−0.51 **	0.13 ^ns^	−0.13 ^ns^	0.34	0.48
GY/P	−9.7 **	9.7 **	5.0 **	−5.0 **	14.4 **	−14.4 **	−7.0 **	7.0 **	−2.7 *	2.7 *	1.9	2.7
AL	−0.015 ^ns^	0.015 ^ns^	−0.025 ^ns^	0.025 ^ns^	0.000 ^ns^	0.000 ^ns^	0.000 ^ns^	0.000 ^ns^	0.040 ^ns^	−0.04 ^ns^	0.167	0.236
AB	−0.013 ^ns^	0.013 ^ns^	−0.003 ^ns^	0.003 ^ns^	−0.003 ^ns^	0.003 ^ns^	−0.008 ^ns^	0.008 ^ns^	0.027 *	−0.027 *	0.02	0.029
FL	0.001 ^ns^	−0.001 ^ns^	0.001 ^ns^	−0.001 ^ns^	−0.004 ^ns^	0.004 ^ns^	−0.014 ^ns^	0.014 ^ns^	0.016 ^ns^	−0.016 ^ns^	0.231	0.327
GL	0.12 *	−0.12 *	−0.11 *	0.11 *	−0.06 ^ns^	0.06 ^ns^	0.07 ^ns^	−0.07 ^ns^	−0.02 ^ns^	0.02 ^ns^	0.1	0.14
GW	0.02 ^ns^	−0.02 ^ns^	−0.03 ^ns^	0.03 ^ns^	0.01 ^ns^	−0.01 ^ns^	−0.06 **	0.063 **	0.059 *	−0.06 *	0.04	0.06
GS	0.02 ^ns^	−0.02 ^ns^	−0.01 ^ns^	0.01 ^ns^	−0.02 ^ns^	0.02 ^ns^	0.07 *	−0.074 *	−0.068 *	0.07 *	0.05	0.07
KL	−0.05 ^ns^	0.05 ^ns^	−0.03 ^ns^	0.03 ^ns^	0.05 ^ns^	−0.05 ^ns^	0.04 ^ns^	−0.04 ^ns^	−0.01 ^ns^	0.01 ^ns^	0.06	0.08
KW	0.036 ^ns^	−0.036 ^ns^	−0.019 ^ns^	0.019 ^ns^	0.003 ^ns^	−0.003 ^ns^	−0.026 ^ns^	0.026 ^ns^	0.006 ^ns^	−0.006 ^ns^	0.041	0.057
KS	−0.041 ^ns^	0.041 ^ns^	0.003 ^ns^	−0.003 ^ns^	0.023 ^ns^	−0.023 ^ns^	0.033 ^ns^	−0.033 ^ns^	−0.017 ^ns^	0.017ns	0.042	0.059
GE	−8.10 **	8.10 **	8.60 **	−8.59 **	−8.59 **	8.59 **	−3.61 **	3.61 **	11.70 **	−11.70 **	1.28	1.82
H (%)	−0.38 ^ns^	0.38 ^ns^	1.15 **	−1.15 **	−0.68 *	0.68 *	0.15 ^ns^	−0.15 ^ns^	−0.25 ^ns^	0.25 ^ns^	0.53	0.75
M (%)	−0.63 *	0.63 *	2.21 **	−2.21 **	−1.26 **	1.26 **	0.81 *	−0.81 *	−1.13 **	1.13 **	0.58	0.82
HR (%)	−4.94 **	4.94 **	0.39 ^ns^	−0.39 ^ns^	1.53 **	−1.53 **	5.06 **	−5.06 **	−2.04 **	2.04 **	1.07	1.51

**: Highly significant at 1% *: Significant at 5% ^ns^: Nonsignificant. PF: Pollen-fertility percentage, SF: Spikelet-fertility percentage, HD: Days to 50% heading (day), PH: Plant height (cm), PL: Panicle length (cm), P/P: Number of panicles per plant, Sp/P: Number of spikelets per panicle, FG/P: Number of filled grains per panicle, PW: Panicle weight (g), 1000-GW: 1000-grain weight (g), GY/P: Grain yield per plant (g), AL: Anther length (mm), AB: Anther breadth (mm), FL: Filament length (mm), GL: Grain length, GW: Grain width, GS: Grain shape, KL: Kernel length, KW: Kernel width, KS: Kernel shape, GE: Grain elongation, H: Hulling percentage, M: Milling percentage and HR: Head rice-recovery percentage.

**Table 7 genes-13-00808-t007:** Percent contribution of different components (lines, testers, and lines × testers) towards the crosses’ sum of squares for various traits in rice.

Traits	Contribution of Line (%)	Contribution of Tester (%)	Contribution of Line × Tester (%)
Pollen fertility (%)	6.01	61.5	32.49
Spikelet fertility (%)	0.58	52.16	47.26
Days to heading	31.74	20.37	47.88
Plant height (cm)	66.09	16.37	17.54
No. of panicles/plant	10.59	14.92	74.49
Panicle length (cm)	58.96	18.85	22.18
No. of spikelets/panicle	56.37	13.41	30.22
No. of filled grains/panicle	55.87	19	25.13
Panicle weight (g)	47.6	43.75	8.66
1000 grain weight (g)	14.39	79.58	6.03
Grain yield/plant (g)	14.6	45.48	39.92
Anther length	0.56	98.75	0.69
Anther breadth	8.39	80.19	11.42
Filament length	0.78	99.2	0.01
Grain length	61.37	32.14	6.49
Grain width	32.2	30.28	37.52
Grain types	72.31	18.37	9.32
Kernel length	26.86	60.89	12.24
Kernel width	62.71	34.17	3.12
Kernel types	67.11	27.9	4.98
Grain elongation	0.51	47.5	52
Hulling (%)	29.01	31.6	39.39
Milling (%)	18.96	52.98	28.05
Head rice (%)	55.75	12.64	31.61

## Data Availability

Not applicable.

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
