# Peer review of "Development of New Iso-Cytoplasmic Rice-Restorer Lines and New Rice Hybrids with Superior Grain Yield and Grain Quality Characteristics by Utilizing Restorers’ Fertility Genes"

_genes, 2022, doi:10.3390/genes13050808_

Round 1

Reviewer 1 Report

In the Materials and Methods section, there is no information on the resulting generations since F1 in 2015 (I1F2, I2F2...). The first three generations of selfing were probably still segregating populations, not homozygous lines. These genotypes may not yet have developed inbreeding depression of traits. For example, in Table 2, four new lines NRL 79, 80, 81 and especially 82 have very high seed productivity per plant (GY/P up to 68.5 g), which is higher than most hybrids. This requires clarification.

In addition, the text of the article and tables often mistakenly say “hidrid check” instead of hybrid check.

Author Response

The attached file is the response to reviewer 1

Reviewer 2 Report

  1. Please add more details and references about the heat map analysis method. There are some questions need to answer:

         1.1 In figures 6 and 7, how to normalize those traits? Different traits have different measuring unit. In general, the heat map is used to visualize expression level of each sample within the same trait ( DOI: 10.3390/genes10120980).

        1.2 In line 322. Hierarchical clustering is a simple but proven method for analyzing gene expression data by building clusters of genes with similar patterns of expression. Thus, it cannot reflect the genetic relationship of the genotypes.  

     2. More detailsand referencesabout the PCA analysis method. Fig 6A and Fig 7A are misused. In order to confirm the conclusion, it is sample (biological replicaiton) not trait that would be clustered into distinct groups (https://doi.org/10.1186/s12870-020-02603-0).

     3. The manuscriptneeds extensive revision for language and grammar.

      3.1 A lot of serious grammatical mistakes in the article should be revised carefully. For example, line 55, 79, 75, 111, 133, 138, 162 and so on.

      3.2 The article is too long and needs to be drastically curtailed. For example, the content in lines 313-315 is repeated.

      3.3 The full name of traits need to be abbreviated. Abbreviations should be marked with the full name when they first appear

  1. Please add the citation in line 50.
  2. The measuring unitof trait should be added in Table 3. 

Author Response

The attached file is the response to reviewer 2.
